# Effect of Dietary *Moringa oleifera* Leaves Nanoparticles on Growth Performance, Physiological, Immunological Responses, and Liver Antioxidant Biomarkers in Nile tilapia (*Oreochromis niloticus*) against Zinc Oxide Nanoparticles Toxicity

Heba S. Hamed [1,*], Rehab M. Amen [2], Azza H Elelemi [3], Heba H. Mahboub [4,*], Hiam Elabd [5], Abdelfattah M. Abdelfattah [6], Hebatallah Abdel Moniem [7], Marwa A. El-Beltagy [8], Mohamed Alkafafy [9], Engy Mohamed Mohamed Yassin [10] and Ayman K. Ismail [11]

[1] Department of Zoology, Faculty of Women for Arts, Science & Education, Ain Shams University, Cairo 11757, Egypt
[2] Department of Zoology, Faculty of Science, Mansoura University, Mansoura 35516, Egypt
[3] Department of Forensic Medicine and Toxicology, Faculty of Medicine, Suez Canal University, Ismailia 41522, Egypt
[4] Department of Aquatic Animals' Medicine, Faculty of Veterinary Medicine, Zagazig University, Zagazig 44511, Egypt
[5] Department of Aquatic Animals' Medicine, Faculty of Veterinary Medicine, Benha University, Moshtohor, Toukh 13736, Egypt
[6] Department of Clinical Pathology, Faculty of Veterinary Medicine, University of Sadat City, Sadat City 32897, Egypt
[7] Animal Production, Department of Animal Wealth Development, Faculty of Veterinary Medicine, Suez Canal University, Ismailia 41522, Egypt
[8] Biochemistry Department, Faculty of Veterinary Medicine, Suez Canal University, Ismailia 41522, Egypt
[9] Department of Biotechnology, College of Science, Taif University, P.O. Box 11099, Taif 21944, Saudi Arabia
[10] Department of Biochemistry, Faculty of Veterinary Medicine, Zagazig University, Zagazig 44511, Egypt
[11] Department of Forensic Medicine and Toxicology, College of Veterinary Medicine, Suez Canal University, Ismailia 41522, Egypt
* Correspondence: hebasalah84@women.asu.edu.eg (H.S.H.); hhhmb@yahoo.com (H.H.M.)

**Abstract:** The current study addresses the influence of *Moringa oleifera* leaves nanoparticles (MO-NPs) on growth, biochemical, immunological, and hepatic antioxidant alterations induced by zinc oxide nanoparticles toxicity in *Nile tilapia* (*O. niloticus*). Fish (N = 180) were divided into four groups with replicates. The first one was set as a control group and the second group was fed an MO-NPs-enriched diet (2.5 g/kg diet). The third group was exposed to 8 mg/L ZnO-NPs, while the forth group was exposed to 8 mg/L ZnO-NPs and fed on MO-NPs (2.5 g/kg diet) for 2 months. Exposure of *O. niloticus* to 8 mg/L ZnO-NPs induced the following consequences: a sharp decrease in the growth parameters; a marked increment in the biochemical biomarkers (glucose, cortisol, and liver enzymes (ALT, AST, ALP); a significant increase in serum renal products, urea and creatinine, cholesterol, and LDH levels. Nonetheless, the dietary MO-NPs supplementation for 2 months significantly alleviated the ZnO-NPs toxicity and significantly enhanced the growth indices, plus normalizing the physio-biochemical levels in the exposed group to ZnO-NPs toxicity to reach the levels of the control group. The MO-NPs markedly improved hepatic antioxidant biomarkers, MDA, and TAC, while decreasing SOD, CAT, and GSH levels to be near the control values. Moreover, supplemented fish in MO-NPs (2.5 g/kg diet) and exposed to ZnO-NPs provided a remarkable increase in the immune profile (respiratory burst (RB) activity, lysozyme, and total immunoglobulins (IgM)) compared to the ZnO-NPs-intoxicated group. Based on the findings of the study, the exposed *O. niloticus* to ZnO-NPs were immune-antioxidant-depressed, besides showing growth retardation, and physio-biochemical alterations. On the other hand, a supplemented diet with MO-NPs is a novel approach to ameliorate ZnO-NPs toxicity for sustaining aquaculture and correspondingly protecting human health.

**Keywords:** *Moringa oleifera* leaves nanoparticles; ZnO-NPs toxicity; hemato-biochemical; oxidative damage; *Oreochromis niloticus*

## 1. Introduction

Fish is a prime source of high-quality protein, vitamins, and other substantial elements like selenium and iodine, which is not found in meat or other crops [1]. Focusing on cultured fish species worldwide, they are mostly exposed to various toxicants mixed in their aquatic environment owing to intensive aquaculture [2,3], including toxicity with nanomaterials [4]. In spite of the wide and safe applications of nanoparticles [5], even in the aquaculture industry as a dietary and an aqueous supplement [6–9], their toxicity commonly occurs in fish [10]. Among NPs, ZnO-NPs are the third most produced ones following $TiO_2$ and $SiO_2$ [11]. The major sources for the entrance of NPs into the aquatic environment include unintentional sources, anthropogenic sources, and the usage of engineered nanoparticles causing adverse effects on all aquatic organisms [12]. The toxicity by ZnO-NPs can induce bioaccumulation in fish tissues, oxidative stress, and sub-acute toxicity in *Cyprinus carpio* [13–16]. Moreover, they induce oxidative damage in zebrafish and *O. niloticus* [17,18], and hepatic and gill damage in *Oreochromis mossambicus* [19]. Therefore, bioaccumulation of ZnO-NPs in fish fillets induces direct toxicity to all fish consumers, mainly human.

Plant extracts achieve a successful way in the aquaculture industry as dietary supplements in promoting growth and immunity, and against NPs toxicity [20]. Specifically, the plant-based NPs are characterized by their porous nature and easy penetration efficacy to fish cells boosting the immune-antioxidant status [21,22]. Among all herbal plants, *Moringa oleifera* (MO) belongs to the family of Moringaceae, native to India, and it is widely distributed worldwide, especially in African and Asian countries [23,24]. It is commonly known as drumstick tree or horseradish tree (Gopalakrishnan et al., 2016). Every part of *M. oleifera* is a storehouse of substantial nutrients. The extract of MO leaves are rich in potassium; iron; phosphorous; vitamins A, C, E, and D; flavonoids; ascorbic acid; β-carotene; oxidase; and catalase [25–27]. Additionally, they provide 17 times more calcium than milk and are utilized to treat malnutrition. About six spoonful's of leaf powder can meet a woman's daily iron and calcium requirements during pregnancy [27]. The pods are fibrous and are valuable for the therapy of digestive problems and for thwarting colon cancer [28]. In addition, MO has long been used in herbal medicine by Indians and Africans and can be used to cure more than 300 diseases [28]. The presence of phytochemicals in MO leaves makes them a good medicinal agent. MO has an effect on diabetes, where MO scavenges the ROS released from mitochondria, thereby protecting the beta cells and in turn controlling hyperglycemia [29,30].

Moringa with its antioxidants can lessen the reactive oxygen species, thereby protecting the brain from cerebral ischemia [31,32]. MO is used to treat dementia, as it has been shown to be a promoter of spatial memory. Additionally, MO can be used as an anti-neoproliferative agent, thereby inhibiting the growth of cancer cells. Therefore, it can be used as an anticancer agent. MO powder can be used as a substitute for iron tablets, hence as a treatment for anemia [27].

Other uses of MO leaves in medicine are that leaves are utilized for the therapy of various diseases as it has many privileges such as hypotensive, antibiotic, hypocholesterolemic, and anti-inflammatory [33,34]. Previous studies indicate that MO has a hepatoprotective effect [26,35] and has a potent immune-antioxidant activity to protect against toxicity in *O. niloticus* [36]. Thus far, just a few studies have assessed the potential effect of neutraceuticals against ZnO-NPs toxicity [22,37]. Therefore, the current study is a new pioneer trial to assess the promoting effect of plant (moringa leaves)-based NPs against the toxicological impacts induced by ZnO-NPs plus their promising influence on growth, physiological, immunological, hepato-renal, and antioxidant response in *O. niloticus*.

## 2. Materials and Methods

### 2.1. Chemicals

Zinc oxide was purchased from Lab Grade—Sigma Company. It was used to synthesize zinc oxide nanoparticles (ZnO-NPs) at the Faculty of Nanotechnology, Cairo University, Egypt, according to [38]. Fresh leaves from the *Moringa oleifera* (MO) plant were picked up from trees grown on sandy soil in El-Sharkia Governorate, Egypt. The collected leaves were purified, washed with distilled water, and dried at room temperature for 3 weeks, after which the leaves were powdered into a coarse form.

Biochemical kits for cortisol, glucose, lactate dehydrogenase (LDH), total proteins, albumin, aspartate aminotransferase (AST), alanine aminotransferase (ALT), alkaline phosphatase (ALP), cholesterol, urea, creatinine, malondialdehyde (MDA), total antioxidant capacity (TAC), superoxide dismutase (SOD), reduced glutathione (GSH), and catalase (CAT) were purchased from Bio-Diagnostic Co. (Cairo, Egypt).

### 2.2. Synthesis of ZnO Nanoparticles (ZnO-NPs)

ZnO-NPs were synthesized at the Faculty of Nanotechnology, Cairo University, Egypt, using the chemical precipitation method beneath the effect of ultrasound irradiation. In a normal process, zinc acetate dihydrate (Zn $(CH_3COO)_2 \bullet 2H_2O$, Loba Chemie, Mumbai, India) as a precursor, and an ammonia ($NH_3$) solution of 30–33% in an aqueous solution ($NH_4OH$, Advent chembio, Mumbai, India) as a reducing agent, had been used. The ZnO-NPs were produced by means of dissolving the right amount of zinc acetate in 100 mL of deionized water to supply 0.1 M of zinc ions. Subsequently, the zinc ions solution was subjected to ultrasonic wave irradiation using a Hielscher UP400S (four hundred W, 24 kHz, Berlin, Germany) at an amplitude of 79% and a cycle of 0.76 for 5 min at a temperature 40 °C [38].

### 2.3. Synthesis of Moringa oleifera Leaves Nanoparticles

Fresh Moringa oleifera (MO) leaves were harvested from trees grown on sandy soil, from El-Sharkia Governorate, Egypt. Then, the collected leaves were purified, washed with distilled water, and air-dried for 4 weeks to remove residual debris. After that, the dried leaves were powdered into a coarse form as described by Hamed and El-Sayed [26]. Then, MO leaves were synthesized into MO leaves nanoparticles (MO-NPs) by a bottom-up process for the synthesis of nanoparticles using the ball mill method. In a typical synthesis, 5 g of MO presenting a micron size was put in a stainless steel dram of ball mill instrument (photon ball mill model of 210 s, Egypt) with ceramics balls of 0.1 cm diameter and 30 g weight.

### 2.4. Transmission Electron Microscopy (TEM)

For TEM analysis, synthetic nanoparticles of ZnO and MO were added to deionized water and sonication for 15 min using ultrasound prop with a 50 kHz, at the amplitude of 50% and 0.59 of a cycle (Up 400s, Hielscher, Germany). Then, 10 microns of desperation were put onto a carbon-coated copper grid (CCG). Electron micrographs were taken using the JEOL model of the GEM-1010 transmission electron microscope at 70 kV.

### 2.5. Fish Maintenance

A total number of 180 healthy *Nile tilapia*, *Oreochromis niloticus* were obtained from the Central Laboratory of Abbassa Fish Farm, Egypt, with an average body weight of 70 ± 5 g. Fish were kept in 100 L glass aquaria containing de-chlorinated tap water under laboratory circumstances (conductivity 2000 L/cm; pH 6.5; oxygen 88–95% saturation; water temperature 25–26 °C; total hardness 150 mg/L as $CaCO_3$; photoperiod 12:12 light/dark) for 2 weeks for acclimatization. Fish received a commercial pellet diet (3% of body weight per day) and were transferred to a fresh volume of water daily.

### 2.6. Experimental Design

*Nile tilapia* (N = 180) were allocated into four equal groups, each group with three replicates of 15 fish per aquarium, and fed on a commercial pellet diet (3% of body weight), which was given to fish up to satiation twice a day at 08:00 and 13:00 for 2 months. The 1st group was set as the control group. The 2nd group was fed on MO-NPs (2.5 g/kg diet). In the 3rd group, the ZnO-NPs were added to the water at the concentration of 8 mg/L for 2 months and this concentration was chosen according to Goda et al. [39]. The 4th group was exposed to 8 mg/L ZnO-NPs and fed on MO-NPs (2.5 g/kg diet) for 2 months. At the end of the experiment, seven fish per group were collected and anesthetized with 0.02% (*v/v*) benzocaine solution. Blood samples were collected from the caudal vessels of fish through two parts. The first part of the samples was allowed to clot in clean dry centrifuge tubes at room temperature and then, centrifuged at $3000\times g$ at 4 °C for 15 min. Fish blood serum was collected for biochemical analysis. However, the second part of the blood samples was collected using a 1 mL heparinized syringe for the determination of respiratory burst activity. After that, fish were dissected and liver samples were collected for antioxidant assay.

### 2.7. Growth Performance Assessment

At the end of the experiment, all fish in each aquarium were counted and weighed, and the calculations were done as follows:

$$\text{Weight gain} = W_2 - W_1$$

$$\text{Specific growth rate (\%/day)} = 100\,(\text{Ln}\,W_2 - \text{Ln}\,W_1)/T$$

where $W_1$ and $W_2$ are the initial and final weights, and T is the experimental period (months).

$$\text{Feed intake (g feed/fish)} = \text{Total feed consumed/fish number/aquarium}$$

$$\text{Feed conversion ratio (FCR)} = \text{Feed intake/fish weight gain.}$$

$$\text{Fish survival (\%)} = 100\,(\text{number of fish at the end/number of fish at the beginning})$$

### 2.8. Biochemical Assays

Serum ALP was assayed following the colorimetric enzymatic method of Tietz et al. [40]. Both AST and ALT activities were evaluated in accordance with the protocol discussed by Reitman and Frankel [41]. Cholesterol estimation was performed following the method described by Allain et al. [42]. LDH was assayed following the protocol registered by Vassault [43]. Creatinine was evaluated following the method described by Larsen [44]. Urea was assayed following Henry et al. [45]. Serum glucose and cortisol levels were measured using the protocols mentioned by Trinder [46] and Foster and Dunn [47], respectively. Serum total protein and albumin were assayed following the steps designated by Lowry et al. [48] and Doumas et al. [49], respectively. The globulin content of the same sample was counted by subtracting the value of albumin from the value of total protein.

### 2.9. Oxidative Stress Biomarkers in Hepatic Tissues

Liver tissue samples were homogenized in cold phosphate-buffered saline (pH 7.4, 0.1 M) with a glass/Teflon homogenizer, and then centrifuged at $3000\times g$ at 4 °C for 15 min. After centrifugation, supernatants were collected and preserved at 20 °C until analysis. Malondialdehyde (MDA) content was assayed following Mihara and Uchiyama [50]. Superoxide dismutase (SOD) and catalase (CAT) were measured following Nishikimi et al. [51] and Aebi [52], respectively. Reduced glutathione (GSH) and total antioxidant capacity (TAC) values were evaluated after the procedures recorded by Beutler et al. [53], and Koracevic et al. [54], respectively.

### 2.10. Immunological Assay

Respiratory burst (RB) activity was estimated through Nitroblue Tetrazolium dye following Secombes [55], as mentioned by Abdel-Tawwab et al. [56]. Lysozyme (LYZ) was assayed through the turbidimetric method by using *Micrococcus luteus* as the substrate in phosphate buffer (pH, 6.2) following Ellis [57]. Total immunoglobulin (IgM) was estimated using the precipitation of polyethylene glycol of IgM, and subtraction of the values of the initial and final total protein following Siwicki and Anderson [58].

### 2.11. Statistical Analysis

The Kolmogorov–Smirnov test was used to test the normality of distribution of all data and the different treatments' homogeneity of variances was assessed through the Bartlett method. Then, the one-way ANOVA data was used to assess the effects of ZnO-NPs toxicity and dietary MO leaves nanoparticles. Differences between means were checked at the 5% probability level using Duncan's test. All the statistical analyses were carried out using the SPSS program (version 20; SPSS, Richmond, VA, USA), as described by Dytham [59].

## 3. Results

### 3.1. TEM Analysis

TEM images of ZnO and MO nanoparticles illustrate that the particles are spherical to sub-spherical in shape with clear edges without any agglomeration (Figures 1 and 2).

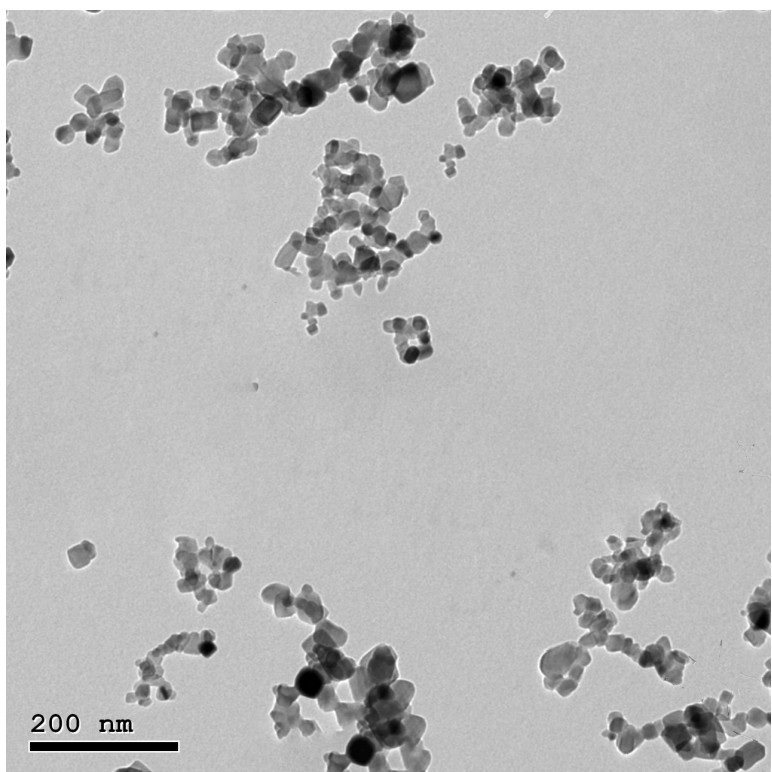

**Figure 1.** TEM image of ZnO-NPs illustrates the spherical and sub-spherical shapes with clear edges without any agglomeration.

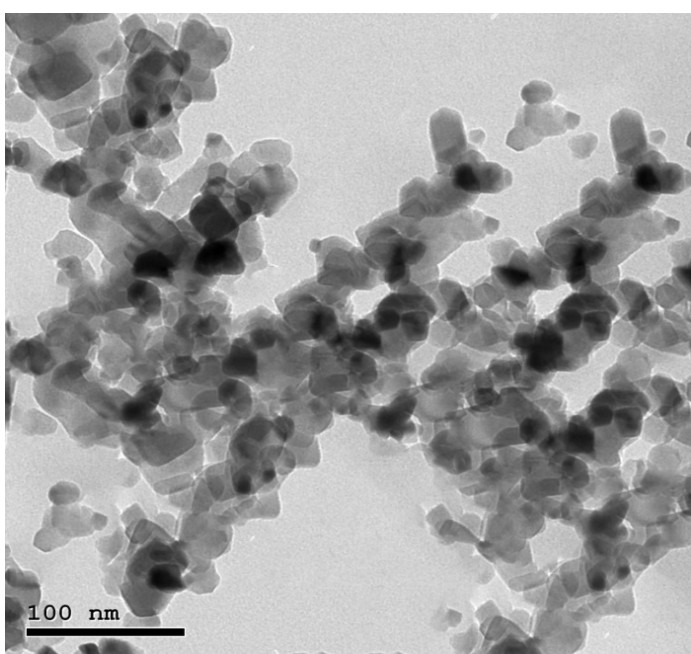

**Figure 2.** TEM image of MO-NPs illustrates the spherical in shape with clear edges without any agglomeration.

*3.2. Growth Parameters*

Dietary MO leaves nanoparticles and ZnO-NPs exposure markedly influenced the growth parameters; however, their interaction was not affected (Table 1; $p < 0.05$). It was noticed that supplementing diets with MO leaves nanoparticles induced significant ($p < 0.05$) increments in comparison with the control group; meanwhile, ZnO-NPs exposure led to marked retardation in FBW, WG%, and SGR (3rd group). Dietary MO leaves nanoparticles significantly alleviated the ZnO-NPs toxicity and significantly enhanced the growth indices. A significant increase in feed intake was recorded with dietary MO leaves nanoparticles; meanwhile, exposing tilapia fish to ZnO-NPs toxicity significantly ($p < 0.05$) depressed the feed intake (Table 1). However, feeding tilapia fish with MO leaves nanoparticles along with ZnO-NPs exposure (4th group) significantly improved the feed intake compared to ZnO-NPs-exposed fish (3rd group).

**Table 1.** Growth and feed utilization indices (mean $\pm$ SD) of *Nile tilapia*, *O. niloticus* exposed to ZnO-NPs with MO leaves nanoparticles administration for 2 months.

| Parameters / Groups | Control | MO-NPs | ZnO-NPS | MO/ZnO-NPS |
|---|---|---|---|---|
| Initial weight (g) | 26.3 $\pm$ 0.44 | 26.19 $\pm$ 0.52 | 26.45 $\pm$ 0.26 | 26.08 $\pm$ 0.41 |
| Final weight (g) | 52.34 $\pm$ 0.42 [b] | 0.32 $\pm$ 57.5 [a] | 37.47 $\pm$ 0.48 [d] | 45.61 $\pm$ 0.23 [c] |
| Weight gain (%) | 112.29 $\pm$ 4.64 [b] | 126.25 $\pm$ 4.31 [a] | 57.62 $\pm$ 2.82 [d] | 77.61 $\pm$ 3.81 [c] |
| SGR (%/day) | 1.42 $\pm$ 0.08 [a] | 1.38 $\pm$ 0.06 [b] | 0.69 $\pm$ 0.02 [d] | 0.98 $\pm$ 0.04 [c] |
| Feed intake (g feed/fish) | 36.7 $\pm$ 3.35 [b] | 43.26 $\pm$ 3.20 [a] | 20.61 $\pm$ 2.29 [d] | 29.42 $\pm$ 2.31 [c] |
| FCR | 1.54 $\pm$ 0.05 | 1.55 $\pm$ 0.03 | 1.58 $\pm$ 0.04 | 1.61 $\pm$ 0.05 |
| Fish survival (%) | 100.0 $\pm$ 1.09 | 100.0 $\pm$ 1.07 | 93.0 $\pm$ 1.0 | 97.0 $\pm$ 1.01 |

Means with different superscript letters in the same row for each parameter are significantly different ($p < 0.05$). Where: Means with different superscript letter in the same row for each parameter are significantly different ($p < 0.05$). MO-NPs group: fish fed diet supplemented with (2.5 g/kg diet) moringa nanoparticles (Mo-NPs), ZnO-NPS group: fish exposed to (8 mg/L) zinc oxide nanoparticles (ZnO-NPs), MO/ZnO-NPS group: fish fed diet supplemented with (2.5 g/kg diet) moringa nanoparticles (Mo-NPs) and exposed to (8 mg/L) zinc oxide nanoparticles (ZnO-NPs). Number of individuals = 7, specific growth rate (SGR), feed conversion ratio (FCR).

### 3.3. Biochemical Analyses

Exposure of tilapia fish to 8 mg/L ZnO-NPs caused marked ($p < 0.05$) increments in levels of serum glucose, cortisol, and liver enzymes (ALT, AST, ALP) compared to the control group (Table 2). Serum renal products, urea and creatinine, cholesterol, and LDH levels were also increased ($p < 0.05$) significantly in ZnO-NPs-exposed fish (Table 3). On the other hand, significant ($p < 0.05$) declines in serum total protein, albumin, and globulin were observed compared to the control tilapia (Table 4). In contrast, feeding tilapia fish intoxicated with ZnO-NPs on a 2.5 g/kg diet of MO leaves nanoparticles for 2 months normalized the levels of the ZnO-NPs profile to be near the levels of the control group. *Nile tilapia*-administered MO leaves nanoparticles exhibited no significant difference in the tested biochemical parameters compared to the control.

**Table 2.** Changes in liver enzymes and stress parameters (mean ± SD) of *Nile tilapia*, *O. niloticus* exposed to ZnO-NPs with MO leaves nanoparticles administration for 2 months.

| Parameters     Groups | Control | MO-NPs | ZnO-NPS | MO/ZnO-NPS |
|---|---|---|---|---|
| AST (U/L) | 50.11 ± 0.24 [b] | 51.01 ± 1.04 [b] | 62.45 ± 1.05 [a] | 50.14 ± 0.13 [b] |
| ALT (U/L) | 43.22 ± 0.41 [b] | 43.67 ± 0.24 [b] | 56.70 ± 0.62 [a] | 42.98 ± 0.07 [b] |
| ALP (U/L) | 9.20 ± 0.17 [b] | 9.26 ± 0.18 [b] | 14.70 ± 0.14 [a] | 9.31 ± 0.29 [b] |
| Glucose (mg/dL) | 47.36 ± 0.11 [c] | 49.01 ± 0.56 [b] | 79.24 ± 1.63 [a] | 50.01 ± 1.61 [b] |
| Cortisol (μg/dL) | 6.13 ± 0.01 [c] | 7.08 ± 0.03 [b] | 12.02 ± 0.11 [a] | 7.18 ± 0.10 [b] |

Where: Means with different superscript letter in the same row for each parameter are significantly different ($p < 0.05$). MO-NPs group: fish fed diet supplemented with (2.5 g/kg diet) moringa nanoparticles (Mo-NPs), ZnO-NPS group: fish exposed to (8 mg/L) zinc oxide nanoparticles (ZnO-NPs), MO/ZnO-NPS group: fish fed diet supplemented with (2.5 g/kg diet) moringa nanoparticles (Mo-NPs) and exposed to (8 mg/L) zinc oxide nanoparticles (ZnO-NPs). Number of individuals = 7, alanine aminotransferase (ALT), aspartate aminotransferase (AST), alkaline phosphatase (ALP).

**Table 3.** Changes in kidney products and lipid profile (mean ± SD) of *Nile tilapia*, *O. niloticus* exposed to ZnO-NPs with MO leaves nanoparticles administration for 2 months.

| Parameters     Groups | Control | MO-NPs | ZnO-NPS | MO/ZnO-NPS |
|---|---|---|---|---|
| Creatinine (mg/dL) | 0.77 ± 0.04 [b] | 0.79 ± 0.02 [b] | 1.45 ± 0.03 [a] | 0.70 ± 0.04 [c] |
| Urea (mg/dL) | 18.34 ± 0.13 [c] | 19.07 ± 0.52 [bc] | 27.61 ± 0.12 [a] | 20.08 ± 0.03 [b] |
| Cholesterol mg/dL | 139.29 ± 3.44 [bc] | 140.51 ± 2.31 [b] | 179.92 ± 1.50 [a] | 140.21 ± 3.21 [b] |
| LDH (U/L) | 20.19 ± 1.18 [c] | 21.02 ± 0.63 [bc] | 37.10 ± 0.62 [a] | 23.51 ± 0.12 [b] |

Means with different superscript letters in the same row for each parameter are significantly different ($p < 0.05$). Means bearing with different subscript are significantly different ($p < 0.05$) as analyzed by one-way ANOVA, MO-NPs group: fish fed diet supplemented with (2.5 g/kg diet) moringa nanoparticles (Mo-NPs), ZnO-NPS group: fish exposed to (8 mg/L) zinc oxide nanoparticles (ZnO-NPs), MO/ZnO-NPS group: fish fed diet supplemented with (2.5 g/kg diet) moringa nanoparticles (Mo-NPs) and exposed to (8 mg/L) zinc oxide nanoparticles (ZnO-NPs). Number of individuals = 7, lactate dehydrogenase (LDH).

**Table 4.** Changes in total proteins (mean ± SD) of *Nile tilapia*, *O. niloticus* exposed to ZnO-NPs with MO leaves nanoparticles administration for 2 months.

| Parameters     Groups | Control | MO-NPs | ZnO-NPS | MO/ZnO-NPS |
|---|---|---|---|---|
| Total protein (g/dL) | 8.22 ± 0.06 [a] | 8.17 ± 0.06 [a] | 4.21 ± 0.06 [c] | 7.58 ± 0.12 [a] |
| Albumin (g/dL) | 4.70 ± 0.03 [a] | 4.81 ± 0.13 [a] | 2.06 ± 0.09 [c] | 4.01 ± 0.30 [b] |
| Globulin (g/dL) | 3.52 ± 0.12 [a] | 3.36 ± 0.15 [b] | 2.15 ± 0.08 [c] | 3.57 ± 0.18 [a] |

Means with different superscript letters in the same row for each parameter are significantly different ($p < 0.05$). MO-NPs group: fish fed diet supplemented with (2.5 g/kg diet) moringa nanoparticles (Mo-NPs), ZnO-NPS group: fish exposed to (8 mg/L) zinc oxide nanoparticles (ZnO-NPs), MO/ZnO-NPS group: fish fed diet supplemented with (2.5 g/kg diet) moringa nanoparticles (Mo-NPs) and exposed to (8 mg/L) zinc oxide nanoparticles (ZnO-NPs). Number of individuals = 7.

### 3.4. Liver MDA and Antioxidant Biomarkers

Nile catfish exposed to 8 mg/L ZnO-NPs exhibited a significant ($p < 0.05$) increase in hepatic MDA, and TAC as presented in Table 5. And markedly decreased ($p < 0.05$) levels of SOD, CAT, and GSH activities of the same tissue compared to the control fish (Table 5). Co-administration of tilapia fish with MO leaves nanoparticles (2nd group) showed no significant difference in hepatic antioxidant biomarkers compared to the control group (1st group). Combined treatment with ZnO-NPs, and MO leaves nanoparticles (2.5 g/kg) resulted in a significant improvement in hepatic antioxidant biomarkers to be near the control values (Table 5).

**Table 5.** Changes in liver antioxidant enzymes and malondialdehyde (MDA) levels (mean $\pm$ SD) of *Nile tilapia*, *O. niloticus*, exposed to ZnO-NPs with MO leaves nanoparticles administration for 2 months.

| Groups / Parameters | Control | MO-NPs | ZnO-NPS | MO/ZnO-NPS |
|---|---|---|---|---|
| SOD (μg/mg protein) | 45.18 ± 0.62 [a] | 44.08 ± 0.43 [b] | 20.70 ± 0.12 [c] | 46.02 ± 0.30 [a] |
| TAC (μmol/mg protein) | 20.19 ± 0.41 [b] | 20.40 ± 0.81 [b] | 35.90 ± 0.03 [a] | 21.05 ± 0.31 [b] |
| CAT (μg/mg protein) | 32.41 ± 0.17 [a] | 33.10 ± 0.10 [a] | 19.04 ± 0.47 [b] | 32.51 ± 0.21 [a] |
| GSH (nmol/g protein) | 23.16 ± 0.13 [a] | 23.12 ± 0.40 [a] | 12.03 ± 0.08 [c] | 20.13 ± 0.62 [b] |
| MDA (nmol/g protein) | 22.15 ± 0.24 [b] | 20.94 ± 0.78 [c] | 37.40 ± 0.93 [a] | 23.43 ± 0.10 [b] |

Means with different superscript letters in the same row for each parameter are significantly different ($p < 0.05$). MO-NPs group: fish fed diet supplemented with (2.5 g/kg diet) moringa nanoparticles (Mo-NPs), ZnO-NPS group: fish exposed to (8 mg/L) zinc oxide nanoparticles (ZnO-NPs), MO/ZnO-NPS group: fish fed diet supplemented with (2.5 g/kg diet) moringa nanoparticles (Mo-NPs) and exposed to (8 mg/L) zinc oxide nanoparticles (ZnO-NPs). Number of individuals = 7, superoxide dismutase (SOD), total antioxidant capacity (TAC), catalase (CAT), reduced glutathione (GSH), malondialdehyde (MDA).

### 3.5. Immunological Assay

Data presented in Table 6 revealed that ZnO-NPs exposed fish (3rd group) experienced marked ($p < 0.05$) reductions in the levels of respiratory burst (RB) activity, lysozyme, and total immunoglobulins (IgM) compared to the controls. In the contrary, combined treatment with ZnO-NPs and MO leaves nanoparticles (2.5 g/kg diet) (4th group) provided a remarkable increase in the immune profile compared to ZnO-NPs-intoxicated group (Table 6).

**Table 6.** Changes in immunological parameters (mean $\pm$ SD) of *Nile tilapia*, *O. niloticus* exposed to ZnO-NPs with MO leaves nanoparticles administration for 2 months.

| Groups / Parameters | Control | MO-NPs | ZnO-NPS | MO/ZnO-NPS |
|---|---|---|---|---|
| LYZ (mg/mL) | 5.82 ± 0.12 [ab] | 6.57 ± 0.24 [a] | 3.71 ± 0.03 [c] | 5.18 ± 0.32 [ab] |
| RBA (mg/mL) | 1.70 ± 0.21 [a] | 1.89 ± 0.34 [a] | 1.13 ± 0.11 [c] | 1.63 ± 0.34 [b] |
| IgM (mg/mL) | 13.53 ± 0.16 [b] | 15.47 ± 0.35 [a] | 8.83 ± 0.21 [d] | 11.94 ± 0.27 [c] |

Means with different superscript letter in the same row for each parameter are significantly different ($p < 0.05$). MO-NPs group: fish fed diet supplemented with (2.5 g/kg diet) moringa nanoparticles (Mo-NPs), ZnO-NPS group: fish exposed to (8 mg/L) zinc oxide nanoparticles (ZnO-NPs), MO/ZnO-NPS group: fish fed diet supplemented with (2.5 g/kg diet) moringa nanoparticles (Mo-NPs) and exposed to (8 mg/L) zinc oxide nanoparticles (ZnO-NPs). Number of individuals = 7, lysozyme (LYZ), respiratory burst activity (RBA), immunoglobulin (IgM).

### 4. Discussion

Nanoparticles toxicity is considered a major challenge for the aquaculture sector. In this regard, the application of neutraceuticals and plant-based NPs has been increasing and represents a significant tool in reaching the goal of sustainable fish health and alleviating toxicity worldwide [60–62] Recent studies utilize different herbals to mitigate the induced toxicity [63–68]. Herein, we are the first report to test the antitoxic role of MO-NPs to

antagonize ZnO-NPs toxicity and to evaluate its promising role as a growth-promoter, an antioxidant-immune supplement, and to enhance fish physiological response.

Blood biomarkers are the key bio-monitoring tools to investigate fish performance and judge the impact of nanoparticles toxicity [63,69]. Herein, we recorded altered hematological profiles in response to ZnO-NPs toxicity. Similar outcomes were obtained by Shahzad et al. [19] that observed the occurrence of DNA damage on erythrocytes of Tilapia fish at a high dose of ZnO-NPs exposure. On the other hand, an enriched diet in MO-NPs retrieved the altered values and caused an elevated level of RBCs reflecting potent antioxidant activity and protection for cell membranes. These outcomes were supported by Hamed and El-Sayed [26] and Sarkar et al. [70] that recorded that the MO protects the RBCs membrane and strengthens its stability. Moreover, Guyton and Hall [71] mentioned that MO contains iron, which is one of the elements for the synthesis of hemoglobin. Concurrent with a recent study, Ibrahim et al. [36] verified that diet rich in MO seeds supplementation improved the blood parameters of *O. niloticus* upon exposure to chlorpyrifos toxicity. Proteins are included in cell architecture and play a substantial role in cell metabolism in fish. The elevated protein content was clear in the MO-NPs-supplemented diet, which reflected better utilization of protein and consequently can enhance growth performance as reported by Glencross et al. [72].

Hepatic enzymes (ALT and AST) are essential enzymes to indicate fish toxicity with NPs [73]. The exposure to ZnO-NPs elevated the key transaminase enzymes, ALT, and AST activity in the blood, reflecting oxidative damage and consequently, hepatotoxicity. These results were confirmed by elevating the antioxidant parameters. These outcomes were in line with a previous report that found elevated levels of hepatic enzymes in *C. carpio* upon exposure to ZnO-NPs toxicity [74]. Nevertheless, fish that received an enriched diet in MO-NPs showed a retrieve in the altered parameters. This could be dominated by the action of kaempferol and quercetin, major components of MO, which enhanced liver function [75]. A similar study by Ibrahim et al. [36] detected amelioration in hepatic enzymes upon the dietary intervention of *O. niloticus* with MO seeds and exposure to chlorpyrifos toxicity.

Based on the findings of the antioxidant assays, it was clear that the group exposed to ZnO-NPs demonstrated a sign of oxidative damage represented in lessening the activity of antioxidant indices (SOD, CAT, GSH, and TAC). This outcome was supported by previous studies that verified that ZnO-NPs were able to induce the production of reactive oxygen species (ROS) [76,77]. The formation of ROS triggers an oxidative damage response because of the direct impact of nanoparticles on cells [78,79], and the excess production of ROS can produce inflammatory and cytotoxic effects [80], and leads to apoptosis [81]. Furthermore, Vimercati et al. [82] elucidated that the mechanism of oxidative stress induced by ZnO-NPs toxicity in freshwater fish is via the generation of ROS, and then damage to proteins, lipids, and DNA, causing cell death. Likewise, Shahzad et al. [19] revealed the occurrence of oxidative stress upon exposure to ZnO-NPs represented in elevated levels of LPO, CAT, GSH, and SOD in the gills and liver of *O. mossambicus*. Moreover, Mahboub et al. [6] reported a state of oxidative damage following exposure of African catfish to ZnO-NPs as represented by a clear increase in oxidative stress parameters (GST, CAT, and GPx). On the other hand, an enriched diet with MO-NPs notably promised the fish antioxidant response via elevating their values reflecting a potent antioxidant action. This could be dominated by the richness of MO in various antioxidants, such as β-carotene, α-and*G*-tocopherol, vitamin A, and β-sitosterol, plus the presence of phenolic constituents, including flavonoids, quercetin, kaempferol, and anthocyanins, as reported by Hamed and El-Sayed [26] and Sánchez-Machado et al. [83]. Additionally, Siddhuraju and Becker [84] documented that MO is rich in vitamin C, which is characterized by many antioxidant properties via its ability to form poorly ionized but soluble complexes with toxic metals. In the same manner, Hamed and El-Sayed [26] and Ibrahim et al. [36] found a strong antioxidant response in *O. niloticus* exerted by MO upon exposure to pendimethalin and chlorpyrifos toxicity, respectively. Furthermore, a recent study by Hamed and Abdel Tawwab [61] utilized

a natural extract, pomegranate peel, to mitigate the oxidative stress induced by silver nanoparticles toxicity in *O. niloticus*.

In the current experiment, the exposure to ZnO-NPs toxicity induced a clear alteration in the immune profile via lessening the measured immune parameters (RBA, lysozyme, and IgM). A recent study by Rashidian et al. [15] supported our findings, as it recorded that exposure to ZnO-NPs toxicity altered the key immunological molecules in both the serum and skin of *C. carpio*. On the other hand, the enriched diet in MO-NPs significantly augmented concentrations of immune biomarkers. A such enhanced immune response could be returned to modifications of cell membranes because of an enriched amount of saturated fatty acids (palmitic and stearic acid) and monounsaturated fatty acids (oleic acid) in MO, as mentioned by DePablo and De Cienfuegos [85]. Another study supported our findings and recorded that MO is rich in vitamins A, C, and K, which strengthen the immune response by stimulating the production of immunoglobulin [23]. Moreover, the presence of amino acids in MO is essential for the formation of immunoglobulin [86]. Concurrently, Ibrahim et al. [36] found enhanced immunity of *O. niloticus* via elevating levels of lysozyme and IgM, following dietary supplement with MO seeds and exposure to chlorpyrifos toxicity.

Taken together, we highlighted for the first time the protective role of dietary MO-NPs as an antidotal natural supplement to combat fish alterations on growth, biochemical, physiological, hepato-renal, and immune-antioxidant induced by ZnO-NPs toxicity in *O. niloticus*.

## 5. Conclusions

The current study demonstrated that ZnO-NPs exposure induces growth suppression, hepato-renal toxicity, protein depletion, immunotoxicity, and oxidative injury in *O. niloticus*. Dietary MO-NPs supplementation can act as a growth promoter and immune-antioxidant stimulator owing to attenuating the ZnO-NPs-induced toxic impacts via stimulating growth, lessening the FCR, enhancing immune-antioxidative response, and modulating hepatic enzymes and protein levels. The present findings highlight the importance of dietary MO-NPs in aquaculture as a tool to enhance fish growth performance and mitigate ZnO-NPs-induced toxicity for both fish and fish consumers. Further studies are much needed to investigate the impact of dietary MO-NPs on gene expression and assess their impacts on other fish species.

**Author Contributions:** Conceptualization; H.S.H., R.M.A., A.H.E., H.H.M., H.E., A.M.A., H.A.M., M.A.E.-B., M.A., E.M.M.Y. and A.K.I.; methodology, H.S.H., R.M.A., A.H.E., H.H.M., H.E., A.M.A., H.A.M., M.A.E.-B., M.A., E.M.M.Y. and A.K.I.; formal analysis, H.S.H., R.M.A., A.H.E., H.H.M., H.E., A.M.A., H.A.M., M.A.E.-B., M.A., E.M.M.Y. and A.K.I.; investigation, H.S.H., R.M.A., A.H.E., H.H.M., H.E., A.M.A., H.A.M., M.A.E.-B., M.A., E.M.M.Y. and A.K.I.; writing—original draft preparation, H.S.H., R.M.A., H.H.M. and H.E.; writing—review and editing, H.H.M. and H.E. All authors have read and agreed to the published version of the manuscript.

**Funding:** Taif University Researchers Supporting Project number (TURSP-2020/57), Taif University, P.O. Box 11099, Taif 21944, Saudi Arabia.

**Institutional Review Board Statement:** The animal study was reviewed and approved by University of Sadat City (approval code VUSC-020-1-22, approved on 7 October 2022).

**Data Availability Statement:** All data are available in this manuscript.

**Acknowledgments:** All the authors are grateful to Taif University Researchers Supporting Project number (TURSP-2020/57), Taif University, P.O. Box 11099, Taif 21944, Saudi Arabia for their support.

**Conflicts of Interest:** The authors declare no conflict of interest.

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
