# Peer review of "Effect of Dietary Moringa oleifera Leaves Nanoparticles on Growth Performance, Physiological, Immunological Responses, and Liver Antioxidant Biomarkers in Nile tilapia (Oreochromis niloticus) against Zinc Oxide Nanoparticles Toxicity"

_fishes, doi:10.3390/fishes7060360_

Round 1
Reviewer 1 Report
The article is an important work on the use of functional additives to mitigate problems in production systems. However, the authors must integrate a characterization of the properties (antioxidants, such as β-carotene, α- andÉ£-tocopherol, vitamin A, and β-317 sitosterol, plus the presence of phenolic constituents, and others) of Moringa oleifera. This would help to understand which compounds are contributing to improving the immune system and mitigating the effects of toxicity by ZnO-NPs.
Address the writing errors in lines 174, 202, and, 203, also in tables 1 and, 2.
Author Response
-The article is an important work on the use of functional additives to mitigate problems in production systems. However, the authors must integrate a characterization of the properties (antioxidants, such as β-carotene, α- andÉ£-tocopherol, vitamin A, and β-317 sitosterol, plus the presence of phenolic constituents, and others) of Moringa oleifera. This would help to understand which compounds are contributing to improving the immune system and mitigating the effects of toxicity by ZnO-NPs.
Response: Thank you for your valuable comment, for clarity, the antioxidants properties of Moringa oleifera were included to better clarify our findings about the ability of Moringa oleifera to improve the immune system and correspondingly, mitigate the effects of toxicity by ZnO-NPs. Please see the added parts in the introduction: It is commonly known as ‘drumstick tree’ or ‘horseradish tree’ (Gopalakrishnan et al., 2016). Every part of M. oleifera is a storehouse of important nutrients and antinutrients. In fact, extracts of MO leaves are rich in potassium, iron, phosphorous, vitamins A, C, E, and D, flavonoids, β-carotene, ascorbic acid, oxidase, and catalase [Gopalakrishnan et al., 2016, Hamed and El- Sayed, 2019], and provide 17 times more calcium than milk, and are used to treat malnutrition, augment breast milk in lactating mothers and about 6 spoonfuls of leaf powder can meet a woman's daily iron and calcium requirements, during pregnancy (Gopalakrishnan et al., 2016). The pods are fibrous and are valuable to treat digestive problems and thwart colon cancer [Oduro et al., 2008 ,Oduro et al., 2008 ]. Also, MO has long been used in herbal medicine by Indians and Africans and can be used to cure more than 300 diseases (Gopalakrishnan et al., 2016). The presence of phytochemicals in MO leaves make it a good medicinal agent. MO has an effect on diabetes. Where, MO scavenges the ROS released from mitochondria, thereby protecting the beta cells and in turn keeping hyperglycemia under control [Kamalakkannan and Prince, 2006, Al- Malki and El- Rabey, 2015].
Moringa with its antioxidants can reduce the reactive oxygen species, thereby protecting the brain from cerebral ischemia [Baker et al., 1998, Kirisattayakul et al., 2013]. MO is used to treat dementia, as it has been shown to be a promoter of spatial memory. Also, MO can be used as an anti-neoproliferative agent, thereby inhibiting the growth of cancer cells. So, it can be used as an anticancer agent. MO powder can be used as a substitute for iron tablets, hence as a treatment for anaemia (Gopalakrishnan et al., 2016).
Reviewer 2 Report
This manuscript reports results of a feeding experiment in tilapia fed a diet enriched with Moringa oleifera leafs. The authors claim that nanoparticles from Moringa oleifera leafs added to the diet of tilapia enhanced growth of the fish and protected tilapia from toxic effects induced by Zink-Oxide nanoparticles in the water. The manuscript is interesting and the data presented are valid.
Before publication, the following points should be improved:
This manuscript describes results of a feeding experiment of tilapia with powedered leaves from Moringa oleifera, the horseraddish tree. The leaves are used in Africa and Asia as food items and have some medicinal importance. Also, it was found to increase for instance the milk yield in cattle. Therefore, the use of this plant in fish feed is justifed, but the background given above is badly described.
Further comments:
Introduction, line 60 ff: The authors claim that fish in aquaculture are frequently exposed to nano-particles. From which sources? This statement should be substantiated. Also the statement, that fish are frequently exposed to ZnO- nano-particles. To me, the rationale of this study appears quite artificial.
Line 85: Moringa oleifera is used in the therapy of various diseases – what kind of diseases? How is this plant used and which parts – leaves, seeds, oil?,
Preparation of Moringa oleifera nano particles, line 101/102: How were the leaves “powdered”? Was the size of the particles measured after “powdering”? How do the authors know that the used “nanoparticles”?
Experimental design:
Line 144, Exposure to ZnO-nanoparticles: Were the nano-particles added to the water? For how long were the fish exposed? Over the entire period of the experiment?
Line 174: Were the fish killed and dissected before the livers were sampled?
Immunological assays
Line 183: Determination of respiratory burst activity: How was this done? In line 148 it is said that blood samples were allowed to clot. However, for an analysis of the respiratory burst by the NBT assay the activity of phagocytic cells is needed. Was a blood sample with heparin collected?
Tables:
Please state the number of individuals (n=?)
Please use different columns headings (control, MO-NO, ZnO-NP, MO/ZnO-NP) instead of group 1-4.
Please explain the meaning of the abbreviations (e.g. SGR, FCR, AST etc) in the foot notes of the tables.
Author Response
This manuscript reports results of a feeding experiment in tilapia fed a diet enriched with Moringa oleifera leafs. The authors claim that nanoparticles from Moringa oleifera leafs added to the diet of tilapia enhanced growth of the fish and protected tilapia from toxic effects induced by Zink-Oxide nanoparticles in the water. The manuscript is interesting and the data presented are valid.
Before publication, the following points should be improved:
This manuscript describes results of a feeding experiment of tilapia with powedered leaves from Moringa oleifera, the horseraddish tree. The leaves are used in Africa and Asia as food items and have some medicinal importance. Also, it was found to increase for instance the milk yield in cattle. Therefore, the use of this plant in fish feed is justifed, but the background given above is badly described.
It is commonly known as ‘drumstick tree’ or ‘horseradish tree’ (Gopalakrishnan et al., 2016). Every part of M. oleifera is a storehouse of important nutrients and antinutrients. In fact, extracts of MO leaves are rich in potassium, iron, phosphorous, vitamins A, C, E, and D, flavonoids, β-carotene, ascorbic acid, oxidase, and catalase [Gopalakrishnan et al., 2016, Hamed and El- Sayed, 2019], and provide 17 times more calcium than milk, and are used to treat malnutrition, augment breast milk in lactating mothers and about 6 spoonful of leaf powder can meet a woman's daily iron and calcium requirements, during pregnancy (Gopalakrishnan et al., 2016). The pods are fibrous and are valuable to treat digestive problems and thwart colon cancer [Oduro et al., 2008, Oduro et al., 2008]. Also, MO has long been used in herbal medicine by Indians and Africans and can be used to cure more than 300 diseases (Gopalakrishnan et al., 2016). The presence of phytochemicals in MO leaves make it a good medicinal agent. MO has an effect on diabetes. Where, MO scavenges the ROS released from mitochondria, thereby protecting the beta cells and in turn keeping hyperglycemia under control [Kamalakkannan and Prince, 2006, Al- Malki and El- Rabey 2015].
Moringa with its antioxidants can reduce the reactive oxygen species, thereby protecting the brain from cerebral ischemia [Baker et al., 1998, Kirisattayakul et al., 2013]. MO is used to treat dementia, as it has been shown to be a promoter of spatial memory. Also, MO can be used as an anti-neoproliferative agent, thereby inhibiting the growth of cancer cells. So, it can be used as an anticancer agent. MO powder can be used as a substitute for iron tablets, hence as a treatment for anaemia (Gopalakrishnan et al., 2016).
Further comments:
Introduction, line 60 ff: The authors claim that fish in aquaculture are frequently exposed to nano-particles. From which sources? This statement should be substantiated. Also the statement, that fish are frequently exposed to ZnO- nano-particles. To me, the rationale of this study appears quite artificial.
Response: For clarity, this part was clarified.
Line 85: Moringa oleifera is used in the therapy of various diseases – what kind of diseases? How is this plant used and which parts – leaves, seeds, oil?,
Response: Every part of M. oleifera is a storehouse of important nutrients and antinutrients. The pods are fibrous and are valuable to treat digestive problems and thwart colon cancer [Oduro etal., 2008 ,Oduro et al., 2008 ]. MO has long been used in herbal medicine by Indians and Africans and can be used to cure more than 300 diseases (Gopalakrishnan etal., 2016). The presence of phytochemicals in MO leaves make it a good medicinal agent. MO has an effect on diabetes. Where, it scavenge the ROS released from mitochondria, thereby protecting the beta cells and in turn keeping hyperglycemia under control [ Kamalakkannan and Prince, 2006, Al- Malki and El- Rabey 2015].
Moringa with its antioxidants can reduce the reactive oxygen species, thereby protecting the brain from cerebral ischemia[ Baker et al., 1998, Kirisattayakul et al., 2013]. MO is used to treat dementia, as it has been shown to be a promoter of spatial memory. Also, MO can be used as an anti-neoproliferative agent, thereby inhibiting the growth of cancer cells. So, it can be used as an anticancer agent. MO powder can be used as a substitute for iron tablets, hence as a treatment for anemia (Gopalakrishnan etal., 2016).
Preparation of Moringa oleifera nano particles, line 101/102: How were the leaves “powdered”? Was the size of the particles measured after “powdering”? How do the authors know that the used “nanoparticles”?
Response:
2.3. Synthesis of Moringa oleifera leaves nanoparticles
Fresh Moringa oleifera (MO) leaves were harvested from trees grown on sandy soil, from El-Sharkia Governorate, Egypt,. Then the collected leaves were purified, washed with distilled water and air dried for 4 weeks to remove residual debris, after which the dried leaves were powdered into a coarse form as described by Hamed and El-Sayed [26]. Then (MO) leaves were synthesized into (MO) leaves-nanoparticles (MO-NPs) by a bottom-up process for synthesis nanoparticles using ball mill method. In a typical synthesis, 5g of MO in micron size is put in stainless steel dram of ball mill instruments (photon ball mill model of 210s, Egypt) with ceramics balls of 0.1cm diameter and 30g weight.
Transmission Electron Microscopy (TEM):
For TEM analysis, synthetic nanoparticles of ZnO and MO were added to deionized water and sonication for 15 min using ultrasound prop with a 50 kHz, at amplitude of 50% and 0.59 of a cycle (Up 400s, Hielscher, German). Then, 10 microns of desperation was put onto a carbon-coated copper grid (CCG). Electron micrographs were taken using JEOL model of GEM-1010 transmission electron microscope at 70 kV.
- Results
3.1. TEM analysis
TEM images of ZnO and MO nanoparticles illustrated that the particles are spherical in shape with clear edges without any agglomeration (Figs.1 & 2).
Fig.1: TEM image of ZnO- NPs illustrated the spherical in shape with clear
edges without any agglomeration
Fig.2: TEM image of MO- NPs illustrated the spherical in shape with clear edges without any agglomeration
Experimental design:
Line 144, Exposure to ZnO-nanoparticles: Were the nano-particles added to the water? For how long were the fish exposed? Over the entire period of the experiment?
Response: The 3rd group, ZnO-NPs were added into water at the concentration of 8 mg/L. for 2 months
Line 174: Were the fish killed and dissected before the livers were sampled?
Response: Corrected.
Fish were dissected and Liver samples were collected for antioxidant assay.
Immunological assays
Line 183: Determination of respiratory burst activity: How was this done? In line 148 it is said that blood samples were allowed to clot. However, for an analysis of the respiratory burst by the NBT assay the activity of phagocytic cells is needed. Was a blood sample with heparin collected?
Response: Corrected.
At the end of the experiment, seven fish per each group were collected and anaesthetized with 0.02% benzocaine solution. Blood samples were collected from the caudal vessels of fish through two parts: The first part of samples were allowed to clot in clean dry centrifuge tubes at room temperature and then centrifuged at 3000 xg at 4°C for 15 min. Sera were collected for biochemical analysis. However, the second part of blood samples were collected using 1-ml heparinized syringe for determination of respiratory burst activity. After that, fish were dissected and Liver samples were collected for antioxidant assay.
Tables:
Please state the number of individuals (n=?)
Response: Done.
Please use different columns headings (control, MO-NO, ZnO-NP, MO/ZnO-NP) instead of group 1-4.
Response: Corrected
. Table1: Growth and feed utilization indices (mean ± SD) of Nile tilapia, O. niloticus exposed to ZnO-NPs with MO leaves nanoparticles administration for 2 months.
Experimental groups
Parameters |
Control group |
MO-NPs group |
ZnO-NPS group |
MO/ ZnO-NPS group |
Initial weight (g/) |
26.3±0.44 |
26.19±0.52 |
26.45±0.26 |
26.08±0.41 |
Final weight (g/) |
52.34±0.42b |
±0.32a.57.5 |
37.47±0.48d |
45.61±0.23c |
Weight gain (%/) |
112.29±.4.64b |
126.25±4.31a |
57.62±2.82d |
77.61±3.81c |
SGR (% /day) |
1.42±0.08a |
1.38±0.06b |
0.69±0.02d |
0.98±0.04c |
Feed intake (g feed / fish) |
36.7±3.35b |
43.26±3.20a |
20.61±2.29d |
29.42±2.31c |
FCR |
1.54±0.05 |
1.55±0.03 |
1.58±0.04 |
1.61±0.05 |
Fish survival (%) |
100.0±.1.09 |
100.0±1.07 |
93.0±1.0 |
97.0±1.01 |
Where: Means with different superscript letter in the same row for each parameter are significantly different (P < 0.05).
Number of individuals= 7
Specific Growth Rate (SGR)
Feed conversion ratio (FCR)
Table 2. Changes in liver enzymes and stress parameters (mean ± SD) of Nile tilapia, O. niloticus exposed to ZnO-NPs with MO leaves nanoparticles administration for 2 months.
Experimental groups
Parameters |
Control group |
MO-NPs group |
ZnO-NPS group |
MO/ ZnO-NPS group |
AST (U/l) |
50.11±0.24b |
51.01±1.04b |
62.45±1.05a |
50.14±0.13b |
ALT (U/l) |
43.22±0.41b |
43.67±0.24b |
56.70±0.62a |
42.98±0.07b |
ALP (U/l) |
9.20±0.17b |
9.26±0.18b |
14.70±0.14a |
9.31±0.29 b |
Glucose (mg/dl) |
47.36±0.11c |
±0.56b49.01 |
79.24±1.63a |
50.01±1.61b |
Cortisol (µg/dl) |
±0.01c6.13 |
7.08±0.03b |
12.02±0.11a |
7.18±0.10b |
Where: Means with different superscript letter in the same row for each parameter are significantly different (P < 0.05).
Number of individuals= 7
Alanine aminotransferase (ALT)
Aspartate aminotransferase (AST)
Alkaline phosphatase (ALP)
Table 3. Changes in kidney products and lipid profile (mean ± SD) of Nile tilapia, O. niloticus exposed to ZnO-NPs with MO leaves nanoparticles administration for 2 months.
Experimental groups
Parameters |
Control group |
MO-NPs group |
ZnO-NPS group |
MO/ ZnO-NPS group |
Creatinine (mg/dl) |
0.77±0.04b |
0.79±0.02b |
1.45±0.03a |
0.70±0.04c |
Urea (mg/dl) |
18.34±0.13c |
±0.52bc19.07 |
27.61±0.12a |
20.08±0.03b |
Cholesterol (mg/dl) |
139.29±.3.44bc |
140.51±2.31b |
179.92±1.50a |
140.21±3.21b |
LDH (U/l) |
20.19±1.18c |
21.02±0.63bc |
37.10±0.62a |
23.51±0.12b |
Where: Means with different superscript letter in the same row for each parameter are significantly different (P < 0.05).
Number of individuals= 7
Lactate dehydrogenase (LDH).
Table 4. Changes in total proteins (mean ± SD) of Nile tilapia, O. niloticus exposed to ZnO-NPs with MO leaves nanoparticles administration for 2 months.
Experimental groups
Parameters |
Control group |
MO-NPs group |
ZnO-NPS group |
MO/ ZnO-NPS group |
Total protein (g/dl) |
8.22±0.06a |
8.17±0.06a |
4.21±0.06c |
7.58±0.12a |
Albumin (g/dl) |
4.70±0.03a |
4.81±0.13a |
2.06±0.09c |
4.01±0.30b |
Globulin (g/dl) |
3.52±0.12a |
3.36±0.15b |
2.15±0.08c |
3.57±0.18a |
Where: Means with different superscript letter in the same row for each parameter are significantly different (P < 0.05).
Number of individuals= 7
Table 5. Changes in immunological parameters (mean ± SD) of Nile tilapia, O. niloticus exposed to ZnO-NPs with MO leaves nanoparticles administration for 2 months.
Experimental groups
Parameters |
Control group |
MO-NPs group |
ZnO-NPS group |
MO/ ZnO-NPS group |
LYZ (mg/ml) |
5.82±0.12ab |
6.57±0.24a |
3.71±0.03c |
5.18±0.32ab |
RBA (mg/ml) |
1.70±0.21a |
1.89±0.34a |
1.13±0.11c |
1.63±0.34b |
IgM (mg/ml) |
13.53±0.16b |
15.47±0.35a |
8.83±0.21d |
11.94±0.27c |
Where: Means with different superscript letter in the same row for each parameter are significantly different (P < 0.05).
Number of individuals= 7
Lyzozyme (LYZ)
Respiratory burst activity (RBA)
Immunoglobulin (IgM)
Table 6. Changes in liver antioxidant enzymes and malondialdehyde (MDA) levels (mean ± SD) of Nile tilapia, O. niloticus, exposed to ZnO-NPs with MO leaves nanoparticles administration for 2 months.
Experimental groups
Parameters |
Control group |
MO-NPs group |
ZnO-NPS group |
MO/ ZnO-NPS group |
SOD (µg/mg protein) |
45.18±0.62a |
44.08±0.43b |
20.70±0.12c |
46.02±0.30a |
TAC (µmol/mg protein) |
20.19±0.41b |
20.40±0.81b |
35.90±0.03a |
21.05±0.31b |
CAT (µg/mg protein) |
32.41±0.17a |
33.10±0.10a |
19.04±0.47b |
32.51±0.21a |
GSH (nmol/g protein) |
23.16±0.13a |
23.12±0.40a |
12.03±0.08c |
20.13±0.62b |
MDA (nmol/g protein) |
22.15±0.24b |
20.94±0.78c |
37.40±0.93a |
23.43±0.10b |
Where: Means with different superscript letter in the same row for each parameter are significantly different (P < 0.05).
Number of individuals= 7
Superoxide dismutase (SOD)
Total antioxidant capacity (TAC)
Catalase (CAT)
Reduced glutathione (GSH)
Malondialdehyde (MDA)
Please explain the meaning of the abbreviations (e.g. SGR, FCR, AST etc) in the foot notes of the tables.
Response: Corrected.
Reviewer 3 Report
In this study, Moringa oleifera leaves nanoparticles were synthetized to serve as (freshwater) fish diet on aquaculture species, as Nile tilapia (Oreochromis niloticus). Effects associated to dietary composed of the developed nanoparticles were evaluated on fish growth performance, physiological and immunological responses and on relevant liver antioxidant biomarkers upon exposure to zinc oxide nanoparticles. It is recognized the interest and relevance of the subject, as this plant has been praised for its health benefits and its encapsulation in nanoparticles is highly prone to enhance its delivery, as compared to the “free phytoactives” present on its leaves. Manuscript is relatively satisfactorily written, but reading is not easy to follow. Particularly, text segments at the introduction and discussion are often confusing and difficult to understand, as rationale of the statements are unclear. A revision by a native English speaker is highly recommended. References are adequate and support statements. In terms of experimental methodology and results, the manuscript is relatively fine executed, but some relevant details require adjustments. Particularly, Moringa oleifera leaves nanoparticles characterization is lacking. Data on nanoparticles physical-chemical characterization by different analytics (DLS, zeta potential and/or TEM/SEM) is lacking, which is important to understand the hydrodynamic diameter of the particles, their morphology, surface charge and polydispersity and how these might contribute for their behavior as “diet agent”. Statistical analysis require just minor edit. Ethical statement must be added. Discussions and conclusions are relevant and supported by collected data, but as similarly to other text segments, thoughts not always seem to be clearly connected and the message to be transmitted is lost. Figures and Tables are pertinent and detailed. Major questions/concerns were highlighted on enclosed document - please see attached.

Author Response
In this study, Moringa oleifera leaves nanoparticles were synthetized to serve as (freshwater) fish diet on aquaculture species, as Nile tilapia (Oreochromis niloticus). Effects associated to dietary composed of the developed nanoparticles were evaluated on fish growth performance, physiological and immunological responses and on relevant liver antioxidant biomarkers upon exposure to zinc oxide nanoparticles. It is recognized the interest and relevance of the subject, as this plant has been praised for its health benefits and its encapsulation in nanoparticles is highly prone to enhance its delivery, as compared to the “free phytoactives” present on its leaves. Manuscript is relatively satisfactorily written, but reading is not easy to follow. Particularly, text segments at the introduction and discussion are often confusing and difficult to understand, as rationale of the statements are unclear. A revision by a native English speaker is highly recommended. References are adequate and support statements. In terms of experimental methodology and results, the manuscript is relatively fine executed, but some relevant details require adjustments. Particularly, Moringa oleifera leaves nanoparticles characterization is lacking. Data on nanoparticles physical-chemical characterization by different analytics (DLS, zeta potential and/or TEM/SEM) is lacking, which is important to understand the hydrodynamic diameter of the particles, their morphology, surface charge and polydispersity and how these might contribute for their behavior as “diet agent”. Statistical analysis require just minor edit. Ethical statement must be added. Discussions and conclusions are relevant and supported by collected data, but as similarly to other text segments, thoughts not always seem to be clearly connected and the message to be transmitted is lost. Figures and Tables are pertinent and detailed. Major questions/concerns were highlighted on enclosed document - please see attached.
Response: Thank you so much for your valuable comments and we really appreciate your efforts for improving the quality of the paper, for clarity, the paper was revised by a native English speaker as required. The introduction was corrected and some sentences were rephrased. The ethical statement is found at the end of the paper: The animal study was reviewed and approved by University of Sadat City (approval code VUSC-020-1-22, approved on 7-10-2022). Different analyses of nanoparticles were carried out and pictures for NPs characterization were added to better clarify the NPs. In addition, all your corrections were considered and corrected as required.
Round 2
Reviewer 1 Report
the authors attended to the comments, thanks.
Author Response
thanks for your review